# An Investigation into the Densification-Affected Deformation and Fracture in Fused Silica under Contact Sliding

**DOI:** 10.3390/mi13071106

**Published:** 2022-07-14

**Authors:** Changsheng Li, Yushan Ma, Lin Sun, Liangchi Zhang, Chuhan Wu, Jianjun Ding, Duanzhi Duan, Xuepeng Wang, Zhandong Chang

**Affiliations:** 1State Key Laboratory for Manufacturing Systems Engineering, Xi’an Jiaotong University, Xi’an 710049, China; li.changsheng@xjtu.edu.cn (C.L.); dzduan@163.com (D.D.); 2State Key Laboratory of Mechanical System and Vibration, Shanghai Jiaotong University, Shanghai 200240, China; 3Wuzhong Instrument Co., Ltd., Wuzhong 751199, China; mys@wzyb.com.cn (Y.M.); 18729961058@163.com (X.W.); czd@wzyb.com.cn (Z.C.); 4Shenzhen Key Laboratory of Cross-scale Manufacturing Mechanics, Southern University of Science and Technology, Shenzhen 518055, China; zhanglc@sustech.edu.cn; 5SUSTech Institute for Manufacturing Innovation, Southern University of Science and Technology, Shenzhen 518055, China; 6Department of Mechanics and Aerospace Engineering, Southern University of Science and Technology, Shenzhen 518055, China; 7School of Mechanical and Manufacturing Engineering, The University of New South Wales, Sydney, NSW 2052, Australia; chuhan.wu@unsw.edu.au

**Keywords:** sliding contact, fused silica, densification, finite element analysis, cracks

## Abstract

Subsurface damage of fused silica optics is one of the major factors restricting the performance of optical systems. The densification-affected deformation and fracture in fused silica under a sliding contact are investigated in this study, via three-dimensional finite element analysis (FEA). The finite element models of scratching with 70.3° conical and Berkovich indenters are established. A refined elliptical constitutive model is used to consider the influence of densification. The finite element models are experimentally verified by elastic recovery, and theoretically verified by hardness ratio. Results of densification and plastic deformation distributions indicate that the accuracy of existent sliding stress field models may be improved if the spherical/cylindrical yield region is replaced by an ellipsoid/cylindroid, and the embedding of the yield region is considered. The initiation sequence, and the locations and stages of radial, median, and lateral cracks are discussed by analyzing the predicted sliding stress fields. Median and radial cracks along the sliding direction tend to be the first cracks that emerge in the sliding and unloading stages, respectively. They coalesce to form a big median–radial crack that penetrates through the entire yield region. The fracture behavior of fused silica revealed in this study is essential in the low-damage machining of fused silica optics.

## 1. Introduction

Fused silica, or silica-rich glass optics, are widely used in laser nuclear fusion devices [1], large astronomical telescopes [2], semiconductor technology [3,4], and consumer electronics. Subsurface damage has plenty of negative effects on the performance of optics, e.g., increasing optical scatter, reducing mechanical strength, and increasing laser-induced damage (LID). For instance, the subsurface damage is one of the precursors resulting in LID. The LID of fused silica optics is one of the key factors restricting the output power and a key challenge for the long-term and stable operation of high-power laser facilities [5]. Therefore, an in-depth understanding of the material removal and damage formation mechanisms of fused silica subjected to machining is essential to fabricate damage-free optics.

Fused silica optics are generally fabricated by abrasive grain-based methods, e.g., grinding and polishing. Therefore, indentation/scratching mechanics are widely used to study the fracture of fused silica subjected to machining [6,7]. In addition to the contact pressure between the indenter and the sample, the indentation stress field also results from a misfit between the plastic zone beneath the indentation and the surrounding elastic matrix. Therefore, the elastic indentation models, e.g., the classical Boussinesq solution and Hertzian field [8] for elastic contact, are insufficient. Later on, Hill et al. [9] developed a model for the wedge indentation of the rigid–perfectly-plastic materials. However, this model is not suitable for indentation with blunt indenters, or with materials with a low ratio of Young’s modulus to yield stress. To this end, Johnson et al. [10] proposed the expanding-cavity model that treats the indentation-induced plastic zone as an expanding zone. This model was successfully used by Lawn et al. [11] to analyze the indentation damage in ceramics.

Different from most materials, fused silica suffered from significant permanent volume contraction under high hydrostatic pressure. This phenomenon is known as densification [12]. The aforementioned indentation stress field models ignore the influence of densification, which limits the accuracy for fused silica. In order to solve this problem, Yoffe [13] proposed the Blister stress field model, which, for the first time, integrates the material densification. Li et al. [14] modified the ECD model to make it suitable for materials with densification. Compared with the Yoffe model, the modified ECD model considers the distribution characteristics of the contact pressure between the indenter and the sample. In addition, the center of the plastic zone is not restricted in the sample surface.

The grinding and polishing processes are more analogous to successive scratching compared with indentations. The studies on analytical sliding stress fields are rather limited. Hamilton and Goodman [15] proposed an elastic model for sliding contact. Ahn et al. [16] developed the sliding blister stress field model by extending the Yoffe model to scratching, in which the plastic deformation and material densification were considered. Similar models were used to analyze the cracking behavior of BK7 glass [17], fused silica [17], and silicon [18] subjected to scratching. However, these models assume that the indenter is conical and the center of plastic zone locates in the sample surface, which limits the prediction accuracy.

The finite element method is a powerful tool to investigate the deformation, friction [19], wear [20], and fracture [21] of brittle materials subjected to scratching. It should be noted, however, that in these studies the constitutive models used were either purely elastic or von Mises [22], and that the effects of material densification were neglected. Imaoka et al. [23] and Gadelrab et al. [24] developed the positive linear models to consider densification. The mean hydrostatic stress is linear with the equivalent shear stress in these models. Xin et al. [25] proposed a negative linear model to explain the unique features of fused silica during grinding and polishing. Kermouche et al. [26] proposed an elliptical constitutive model to consider the shear-assisted densification. This model considers the hardening of yield pressure with densification, which is neglected in the linear models. The elliptical model is widely used to investigate the indentation deformation and fracture in fused silica [12,27]. Later on, the elliptical constitutive model was refined by Li et al. [28], to consider the influence of densification on elastic properties and the saturation of densification with hydrostatic pressure. This refined elliptical model was successfully used to study the indentation mechanisms [28,29] and sliding friction behavior [30].

This paper aims to establish three-dimensional finite element models for conical and Berkovich scratching using the refined elliptical model. Finite element simulations are performed to investigate the densification and deformation in fused silica subjected to scratching to reveal the stress field more precisely. The influence of friction on indentation and scratching hardness is investigated, and the cracking behavior of fused silica under scratching explored.

## 2. Scratching Tests

As shown in Figure 1, fused silica samples (Corning UV 7980, Corning Corp., Corning, NY, USA) were scratched by an edge-forward Berkovich tip on a nanoindentation machine (TI-950 TriboIndenter, Hysitron Inc., Eden Prairie, MN, USA). The samples were carefully polished to achieve a surface roughness small than 2 nm. Scratching tests were performed under constant normal loads of 1 mN, 2 mN, 4 mN, 200 mN, 400 mN, 600 mN, 1 N, and 1.2 N. The sliding length was 250 μm, which is significantly greater than the scratching depth. The scratching process consists of the approaching stage Ⓐ, the preliminary profiling stage Ⓑ (to obtain the original surface profile), the indentation stage Ⓒ, the scratching stage Ⓓ, the unloading stage Ⓔ, and the postmortem profiling stage Ⓕ (to obtain the residual surface profile). The variations of normal load, normal displacement, lateral load, and lateral displacement with time were recorded during scratching.

After scratching, the samples were measured by an atomic force microscope (AFM) (Innova, Veeco, Plainview, NY, USA) to obtain the three-dimensional topography of the impression. After etching by the buffered HF solution to open the surface cracks [31], the morphology of the cracks was characterized by a scanning electron microscope (SEM) (SU3500, Hitachi, Japan).

## 3. Finite Element Modeling

The finite element analysis of scratching with an edge-forward Berkovich indenter and a conical indenter was performed on a commercial finite element code ABAQUS. A modified elliptical constitutive model [28,30] was used to consider the influence of densification on the deformation in fused silica:(1)$$f\left(\sigma_{ij}\right)={\left(\frac{q}{d}\right)}^{2}+{\left(\frac{p}{p_{b}}\right)}^{2}-1=0$$
where *q* is equivalent shear stress; *p* is hydrostatic pressure; *d* is the von Mises yield stress under pure shear; and *p_b_* is the hydrostatic yield stress for pure compression. The relationship between hydrostatic pressure *p* and the densification ζ is modeled by:(2)$$\zeta =\frac{\zeta_{\max}}{1+e^{-k\left(p-p_{0}\right)}}$$
where $\zeta_{\max}$ (%) is the saturated densification under compression, and *p*_0_ (GPa) is the hydrostatic pressure under which a densification of $\zeta_{\max}$/2 is produced. The parameters of the modified elliptical model used in this study are taken from the ref. [28].

In the finite element model, an infinitely sharp Berkovich edge-forward indenter slides along the *x*-axis on the top surface of a deformable parallelepiped with a dimension of *W* × *W* × (*l* + 2*W*), as shown in Figure 2. The diamond indenter is assumed to be rigid because its Young’s modulus [32] and hardness [33] are much higher than those of the fused silica samples [14]. An eight-node linear brick element with reduced integration and hourglass control is used. Refined FE mesh with an element size of *l_e_* is used in a parallelepiped with a dimension of *a* × *a* × (*l* + 2*W*), and graded FE mesh is used in the residual region. For conical scratching, the semi-included angle *α* of the conical indenter is set as 70.3°, to ensure that the projected area-to-indentation depth function is the same as the commonly used Vickers and Berkovich indenters. *A* = 2.79 *h_max_* is the nominal contact radius for 70.3° conical scratching.

As shown in Figure 2 and Figure 3, the sliding process is divided into three stages, i.e., the indentation stage ①, the sliding stage ②, and the unloading stage ③. The scratching depth, length, and speed are denoted as *h_max_*, *l,* and *v*, respectively. The Coulomb friction model is used to model the adhesion friction behavior between the indenter and the sample. The coefficient of adhesion friction *f* was determined to be 0.04, by comparing FEA and scratching tests.

The single-variable method is used to optimize the cross-section dimension *W*, sliding length *l,* and element size *l_e_*. The appropriate parameters result in stable and convergent normal and tangential loads, and apparent coefficient of friction in the sliding stage. *h_max_* is assumed to be 1 μm. Results show that a cross-section dimension of 5*a* × 5*a*, a sliding length of 10 *h_max_*, and a mesh size of 1/8 *h_max_* are appropriate for the simulations.

## 4. Verification of Finite Element Models

### 4.1. Experimental Verification of Elastic Recovery

The elastic recovery ratio *f_e_* for scratching reflects the extent of elastic deformation relative to the whole deformation. In addition, *f_e_* can be conveniently measured by AFM. Therefore, *f_e_* is used to verify the finite element model in this study.

As shown in Figure 4, the leading end of the impression induced by scratching with an edge-forward Berkovich indenter is measured by AFM to obtain its three-dimensional topography. Pile-up is obvious on the lateral sides of the impression. Figure 4 also indicates that the residual depth *h_f_* (see Figure 5) slightly decreases with the distance *d* to the unloading position of indenter tip. The profile shown in Figure 5 is obtained by averaging five equally spaced cross-section profiles of the middle part of the impression. The residual scratch depth *h_f_* after elastic recovery, determined from Figure 5, is 668 nm.

In order to determine the scratching depth directly from the displacement curve, the normal displacement of the indenter (see Figure 1c) during the scratching process was corrected by the original profile of the sample surface. The corrected normal displacement in the scratching stage Ⓓ was calculated by subtracting the uncorrected displacement from *t*_2_ to *t*_1_ in the stage Ⓑ from the uncorrected displacement from *t*_3_ to *t*_4_. Similarly, the corrected normal displacement in the postmortem profiling stage Ⓕ was calculated by subtracting the uncorrected displacement from *t*_1_ to *t*_2_ from the uncorrected displacement from *t*_5_ to *t*_6_. The evolution of the corrected normal displacement with time is shown in Figure 6. The maximum scratching depth and residual depth are 1063 nm and 479 nm, respectively. It is worth noting that this value of residual depth is smaller than that measure by AFM (i.e., *h_f_* = 668 nm). This is possibly because the indenter did not strictly follow the scratching path in stage Ⓕ, due to the movement of the sample or the motion error of the indentation test in the lateral direction. By contrast, the AFM probe accurately detects the lowest positions of the residual scratching profiles for two reasons. First, the tip radius of the AFM probe is much smaller than that of the Berkovich indenter. Second, the AFM probe is scanning across the impression. Using the AFM-measured *h_f_*, the elastic recovery ratio is calculated to be *f_e_* = 1−*h_f_*/*h_amx_* = 37.2%. Ba analyzing the FEA-predicted profiles of the scratching impression at the fully-loaded and fully-unloaded states shown in Figure 7, the value of *f_e_* predicted by FEA is 37%, which is very close to the experimental value.

### 4.2. Theoretical Verification of Hardness Ratio

As shown in Figure 8, FEA predicts that the hardness ratio $k_{H}^{p}=H_{T}^{p}/H_{s}^{p}$ is slightly larger than utility, and nearly independent of the sharpness of the indenter, where $H_{T}^{p}$ and $H_{s}^{p}$ are the ploughing hardness along the sliding and vertical directions, respectively. This is consistent with theoretical analysis, as detailed below.

For an infinitesimal contact area *dA*, the contact force is normal to *dA*. Therefore, the forces along the sliding and vertical directions, i.e., *dF_T_* and *dF_N_*, have the following relationship:(3)$$\frac{dF_{T}}{dA_{pl}}=\frac{dF_{N}}{dA_{pv}}=p(h,\beta )$$
where *A_pl_* and *A_pv_* are the laterally and vertically projected contact areas, respectively; *p*(*h*,*β*) is the contact pressure at the point (*h*,*β*) on the indenter surface; and *β* and *h* are the phase and the height measured from the indenter tip, respectively. According to the definition of hardness, the ploughing hardness can be expressed by:(4)$$H_{s}^{p}=\frac{\int p\left(h,\beta \right)dA_{pv}}{A_{pv}}$$
and
(5)$$H_{T}^{p}=\frac{\int p\left(h,\beta \right)dA_{pl}}{A_{pl}}$$

The contours of the contact stress induced by conical scratching and Berkovich scratching resemble concentric circles and triangles, respectively [30]. Therefore, *p*(*h*,*β*) is nearly independent of *β*. As the geometries of the conical and Berkovich indenters are self-similar, the vertically and laterally projected areas of the contact zone with a height from *h* to *h* + *dh*, i.e., $dA_{pl,h∼h+\Delta h}$ and $dA_{pv,h∼h+\Delta h}$ have the following relationship:(6)$$\frac{dA_{pl,h∼h+\Delta h}}{dA_{pv,h∼h+\Delta h}}\approx \frac{A_{pl}}{A_{pv}}$$

Therefore, the hardness ratio:(7)$$k_{H}^{p}=\frac{H_{T}^{p}}{H_{s}^{p}}=\frac{\int_{h=0}^{h_{c}}p\left(h\right)dA_{pl,h∼h+\Delta h}}{\int_{h=0}^{h_{c}}p\left(h\right)dA_{pv,h∼h+\Delta h}}\cdot \frac{A_{pv}}{A_{pl}}\approx 1$$

By combining Equations (3), (12) and (18) in ref. [30], the following expression can be obtained for a conical indenter:(8)$$\frac{k_{H}^{}}{k_{H}^{p}}\approx 1+\frac{f}{\mu_{0}\sin\alpha }$$
where *k_H_* = *H_T_*/*H_s_* is the ratio of tangential hardness and scratching hardness; and $\mu_{0}$ is the friction coefficient induced by ploughing. As $k_{H}^{p}\approx 1$, we can conclude from Equation (8) that *k_H_* > 1 when friction exists, i.e., *H_T_* > *H_s_*. This is consistent with the scratching tests [34]. It is worth noting that the above analysis considers the non-uniform distribution of the contact pressure. This is more accurate than the widely adopted assumption that the contact pressure is uniformly distributed [35].

## 5. Deformation and Fracture in Fused Silica under Scratching

### 5.1. Scratching Hardness

Scratching hardness is widely used to model the scratching load that is a key factor determining the fracture behavior. Although it is reported that friction only plays a small role in indentation hardness for blunt indenters [36], Figure 9 shows that the indentation hardness *H_i_* (the hardness at the end of stage ①) for edge-leading Berkovich scratching is slightly increased with the rise in friction. By contrast, the scratching hardness (the hardness in the right red box) is nearly independent of friction.

In order to understand the above phenomena, the evolutions of the contact area and normal force with time are plotted in Figure 10a,b, respectively. During the transition from indentation to sliding, i.e., in the stage ②-1, the contact area is considerably reduced because the support for the indenter rear is removed. If no elastic recovery occurs, the rear face of the indenter is entirely separated from the sample surface, and the contact area decreases to two-thirds. However, Figure 10a shows that the contact area during the steady sliding stage (in the right red box) is significantly bigger than two-thirds of that during indentation (in the left red box). This is due to elastic recovery.

The moving direction of the indenter relative to the sample determines the influence of friction on the contact area. The contact area during static indentation is slightly decreased by friction, because the friction-induced downward shear stress applied to the sample surface results in a decrease in the contact area, as shown in Figure 10a. By contrast, friction leads to an increase in the contact area during the steady sliding stage because the friction-induced shear stress on the sample surface is upward. The indentation hardness increases with friction, while the contact area decreases with friction. Therefore, the normal force during indentation is nearly independent of friction, as shown in Figure 10b. During the steady sliding stage, the scratching hardness is nearly independent of friction because the friction increases both the contact area and the normal force.

Although the scratching hardness *H_s_* for Berkovich indenter is independent of *f*, *H_s_* for conical indenter is linearly decreased with *f*, as demonstrated in Figure 11. The scratching hardness induced by ploughing, i.e., $H_{s}^{p}$, remains nearly unchanged when *f* increases from 0 to 0.2.

### 5.2. Plastic Deformation

It is reported that the stress and deformation are relieved by densification [13,25]. The stress distribution is significantly influenced by the geometry and location of the elastic–plastic boundary [14,37]. In the analytical models of sliding stress fields, e.g., the Ahn model [16] and the Wang model [17], the plastic region is assumed to be a sphere with the center on the sample surface.

As shown in Figure 12, the maximum densification locates in the region that the indenter tip passes. The maximum value of densification predicted by FEA, i.e., 22.6%, is close to the measured saturated densification, i.e., 21% [38]. By comparing the densification contours in the top surface and the *xz*-cross-section shown in Figure 12, it is found that the contours in the *yz*-cross-sections are flat ellipses. Figure 12 also indicates that the densification in fused silica caused by scratching is bigger than that caused by indentation.

Figure 13a shows that the elastic–plastic boundary in fused silica induced by scratching deviates from a circle. By contrast, Figure 13b demonstrates that the boundary can be tightly fitted by an ellipse. Figure 13 and Figure 14 indicate that the yield region is an ellipsoid in the front of the indenter (*x*/*h_max_* > 10), and a cylindroid at the rear of the indenter (*x*/*h_max_* ≤ 10).

As shown in Figure 13b, the shape of the elastic–plastic boundary is characterized by the length *m* of the semi-major axis, the length *n* of the semi-minor axis, and the depth *ξ* of the elastic–plastic boundary center. The semi-major and semi-minor axes are along the lateral and vertical directions, respectively. These parameters can be determined by fitting the elastic–plastic boundary by the following formula:(9)$${\left(\frac{y}{m}\right)}^{2}+{\left(\frac{z-\xi }{n}\right)}^{2}=1$$

The fitted results are shown in Figure 15. The real contact radius *a_r_* predicted by FEA equals 2.24 *h_max_*. Figure 15a shows that *m* is bigger than *a*, while *n* is smaller than *a*. After unloading, *m* remains nearly unchanged, but *n* in the cross-section close to point *B* increases, due to the significant elastic recovery. *m* is significantly larger than *n* in the steady sliding stage. Therefore, the prediction accuracy of the existent sliding stress field models may be greatly improved if the spherical/cylindrical yield region is replaced by an ellipsoid/cylindroid. Figure 15b shows that the depth of the elastic–plastic boundary center in the *yz*-cross-section behind the indenter decreases rapidly when the indenter moves far away from it. *ξ* for scratching is much bigger than that for indentation (close to *ξ* at *l_B_* = 9.5 *h_max_*) at both the fully loaded and the fully unloaded states. This indicates that the Ahn and Wang models should be refined to allow for the embedding of the center of the plastic zone.

### 5.3. Stress Fields and Cracking Behavior

#### 5.3.1. In the Sliding Stage

The scratching-induced stress contours under a conical indenter at the end of the sliding stage ② are shown in Figure 16. In the front of the indenter tip, i.e., in the region *x* ≥ 0, the shape of the contour lines induced by scratching is similar to those induced by indentation [29]. By contrast, at the back of the indenter tip, the contours are flattened, due to the plastic deformation left by the sliding indenter.

The maximal values of *σ_x_* and *σ_y_* are identified in the regions below the indenter tip just outside the elastic–plastic boundary, i.e., the regions *R*_1_ and *R*_2_ shown in Figure 16, respectively. They are the driving forces of median cracks. As the maximal *σ_y_* is higher than the maximal *σ_x_*, the median crack along the sliding direction tends to initiate prior to that along the lateral direction. The maximal value of *σ_z_* locates at the far rear of the indenter (region *R*_3_), which is the driving force of lateral cracks. In the sliding stage, the driving force of median cracks is higher than that of lateral cracks.

#### 5.3.2. At the Fully-Unloaded State

After unloading, as shown in Figure 17a, the stress contours predict a high *σ_y_* on the sample surface at the front of the indenter (region *R*_4_), which is the driving force of radial cracks along the sliding direction, i.e., the radial crack 1 in Figure 18a. *σ_y_* at the bottom of the yield region remains nearly unchanged after unloading. Therefore, median cracks along the sliding direction remain open in the unloading stage if they initiate in the sliding stage. The maximal *σ_z_* increases from 0.087*H* to 0.137*H* during the unloading process. This indicates that the lateral crack emerges more easily in the unloading stage compared with the sliding stage.

#### 5.3.3. Maximum Principal Stress

The contours of *σ*_1_*/H* shown in Figure 19 indicate that the median crack along the sliding direction tends to be the first crack to appear during the sliding stage. It initiates below the yield region at the rear of the indenter, i.e., in the region *R*_2_. Once initiated in the *xz*-cross-section, the median crack propagates along the sliding direction during scratching. As the value of *σ*_1_ in region *R*_3_ is smaller than that in region *R*_2_, the initiation load of median crack under indentation is higher than that under scratching. This is consistent with experimental observations. Though median crack is absent during indentation tests under the normal load of 40 N [39], it is observed during scratching tests under the normal load of 600 mN, as shown in Figure 18b. During the unloading stage, radial cracks may initiate from the sample surface at the front of the indenter, i.e., region *R*_4_, as shown in Figure 18a. As *σ_y_* in region *R*_4_ is small before unloading, and significantly increases during the unloading process, radial cracks tend to emerge in the unloading stage. If both radial and median cracks form, they coalesce to form a big median–radial crack that penetrates through the entire yield region, as verified by the experiments (see Figure 18).

## 6. Conclusions

This study developed three-dimensional finite element models of scratching on fused silica with 70.3° conical and Berkovich indenters. A refined elliptical model was used to consider the influence of densification on elastic properties and the saturation of densification with hydrostatic pressure. By analyzing the predicted scratching hardness, plastic deformation, and stress fields, the following conclusions are obtained:

(1) The tangential hardness is slightly larger than the scratching hardness, and their ratio linearly increases with the adhesion friction coefficient *f*. The scratching hardness for an edge-leading Berkovich indenter is nearly independent of *f*, because the friction increases both the contact area and the normal force. By contrast, the scratching hardness for a conical indenter linearly decreases with *f*. These findings are helpful to model the friction-affected forces induced by scratching;

(2) The densification effect should not be ignored if one aims for a damage-free process in fabrication. It is found that the material’s densification under scratching is bigger than that under indentation. This indicates that the prediction accuracy of the sliding stress analyses can be improved if the material’s densification is properly integrated into the modelling, e.g., replacing the spherical/cylindrical plastic zone with an ellipsoid/cylindroid, and considering the embedding of the plastic zone center;

(3) Median cracks along the sliding direction tend to be the first cracks that emerge in the sliding stage. Radial cracks may initiate under a smaller load in the sample surface at the front of the indenter during the unloading stage. Radial and median cracks coalesce to form a big median–radial crack that penetrates through the entire yield region.

## Figures and Tables

**Figure 1 micromachines-13-01106-f001:**
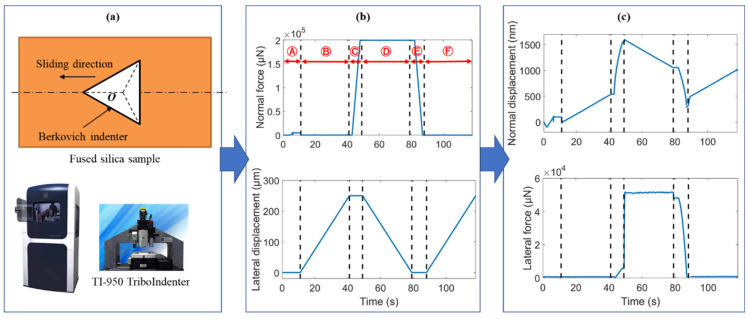
(**a**) Schematic diagram of edge-forward Berkovich scratching, (**b**) the applied normal force and lateral displacement for the normal load of 200 mN, and (**c**) the resulting normal displacement and lateral force.

**Figure 2 micromachines-13-01106-f002:**
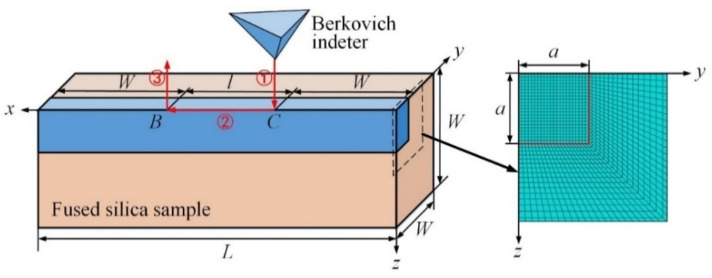
The finite element model of scratching with a Berkovich indenter.

**Figure 3 micromachines-13-01106-f003:**
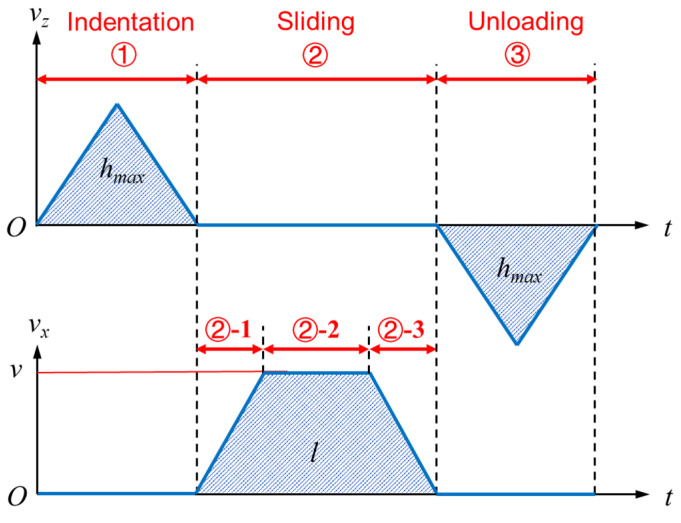
The velocity of the indenter during the FEA of scratching.

**Figure 4 micromachines-13-01106-f004:**
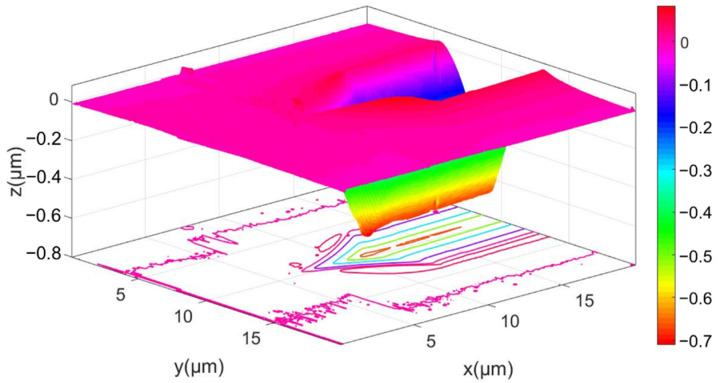
The AFM-measured three-dimensional topography of the scratching impression induced by an edge-forward Berkovich indenter under the normal load of 200 mN.

**Figure 5 micromachines-13-01106-f005:**
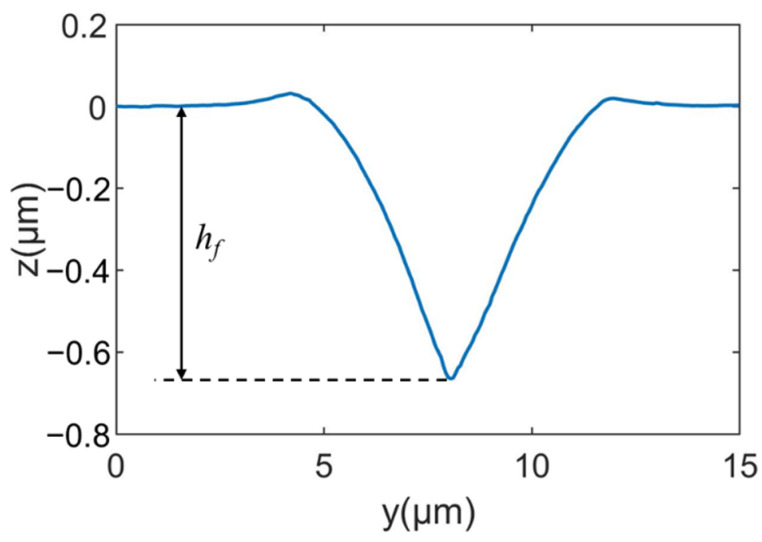
The average cross-section profile of the scratching impression.

**Figure 6 micromachines-13-01106-f006:**
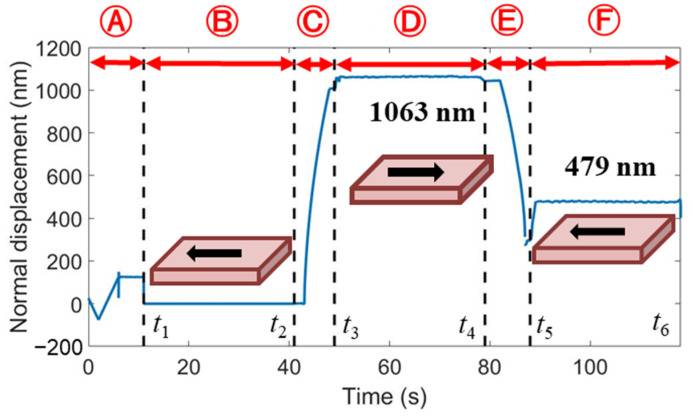
The evolution of the corrected normal displacement with time for scratching with an edge-forward Berkovich indenter, under the normal load of 200 mN.

**Figure 7 micromachines-13-01106-f007:**
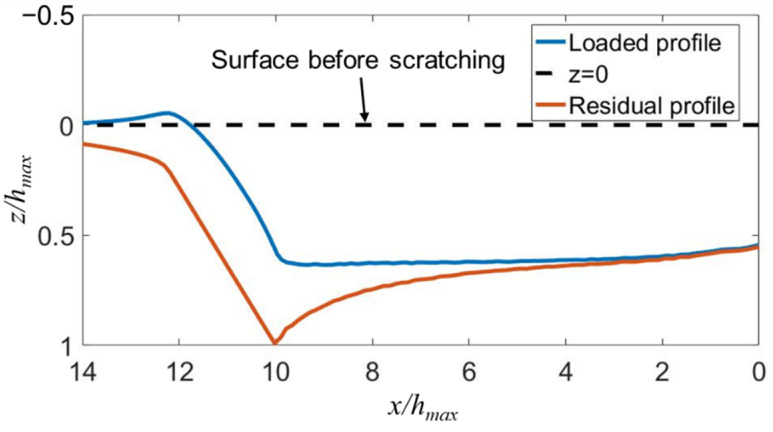
The FEA-simulated profiles of the scratching impression at the fully loaded and fully unloaded states for an edge-forward Berkovich indenter.

**Figure 8 micromachines-13-01106-f008:**
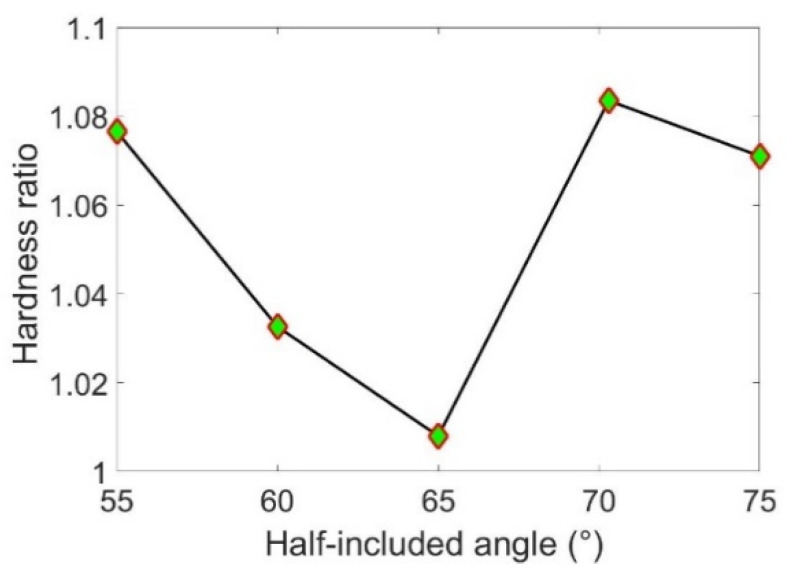
The variation of hardness ratio $k_{H}^{p}$ with the half-included angle of the conical indenters.

**Figure 9 micromachines-13-01106-f009:**
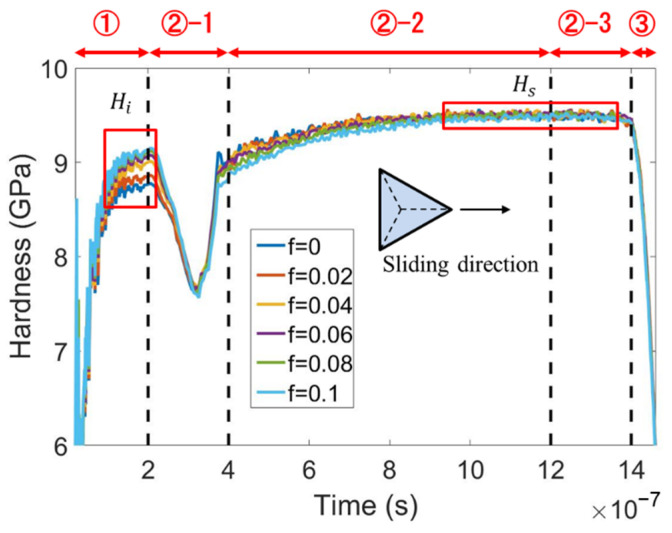
The variation of hardness with time at various values of adhesion friction coefficient *f* for edge-leading Berkovich scratching.

**Figure 10 micromachines-13-01106-f010:**
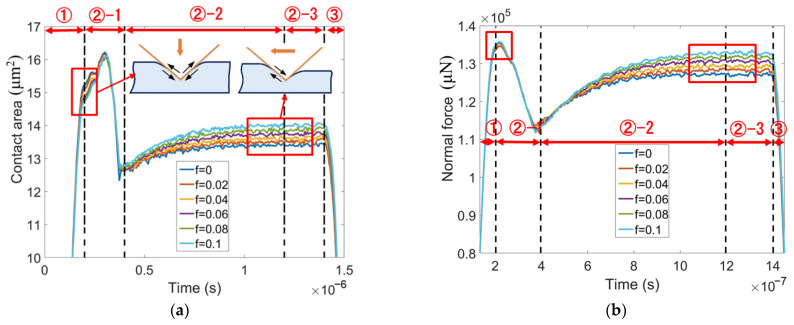
The evolutions of (**a**) contact area and (**b**) normal force with time at various friction coefficients for edge-leading Berkovich scratching.

**Figure 11 micromachines-13-01106-f011:**
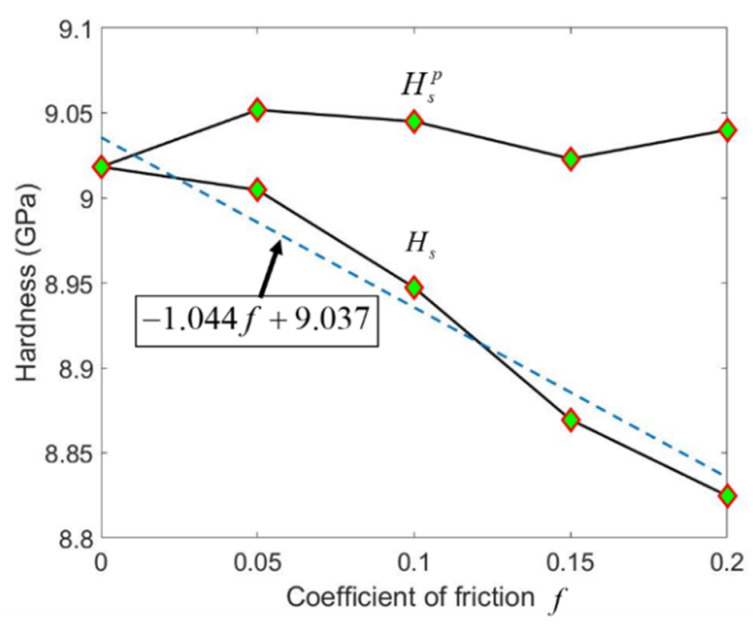
The hardness during scratching as a function of adhesion friction coefficient *f* for scratching with a 70.3° conical indenter. The value pairs of (*H_s_*, *f*) is fitted to obtain the dash line.

**Figure 12 micromachines-13-01106-f012:**
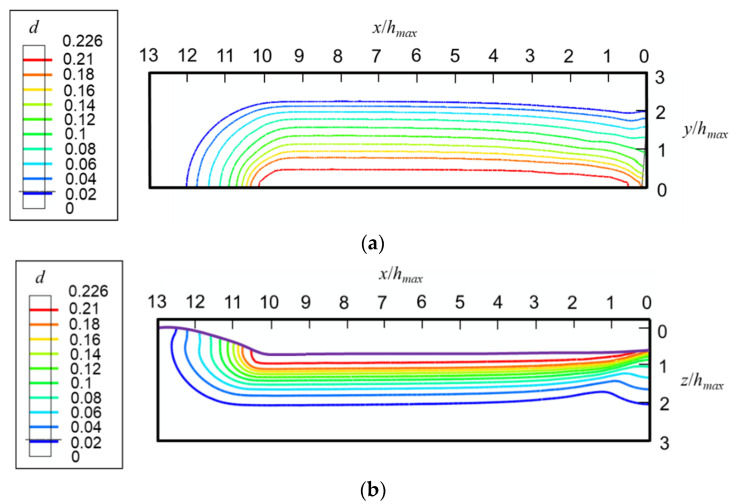
Contours of densification at the fully unloaded state induced by scratching with a 70.3° conical indenter in (**a**) the top surface and (**b**) *xz*-cross-section.

**Figure 13 micromachines-13-01106-f013:**
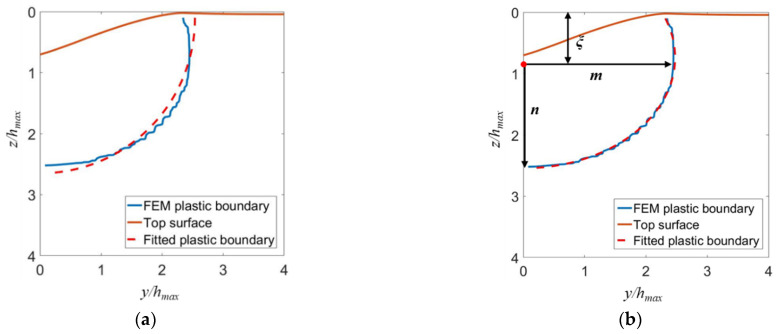
The elastic–plastic boundaries defined by a von Mises equivalent plastic strain of 10^−2^ in the *yz*-cross-section at the fully unloaded state for 70.3° conical scratching. The simulated boundaries are fitted by (**a**) a circular arc and (**b**) an elliptical arc.

**Figure 14 micromachines-13-01106-f014:**
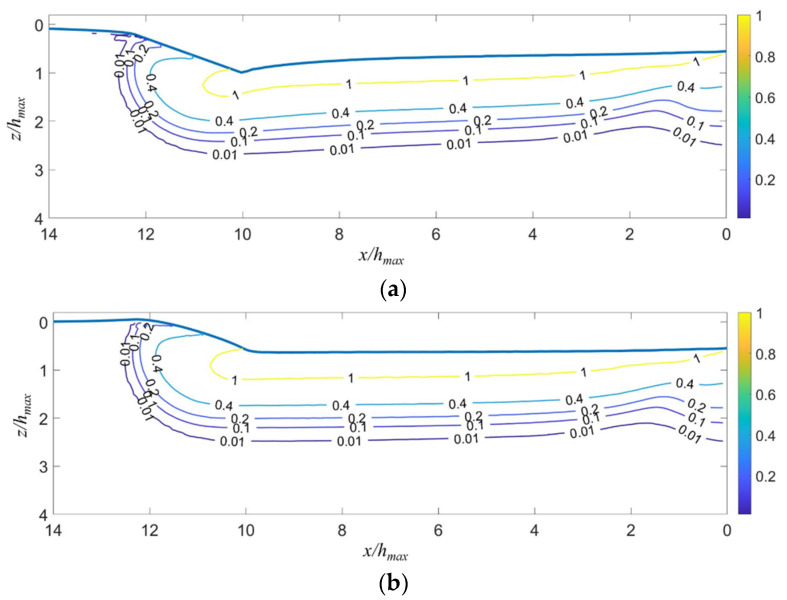
The contours of von Mises equivalent plastic strain in the *xz*-cross-section (**a**) at the fully loaded and (**b**) fully unloaded states.

**Figure 15 micromachines-13-01106-f015:**
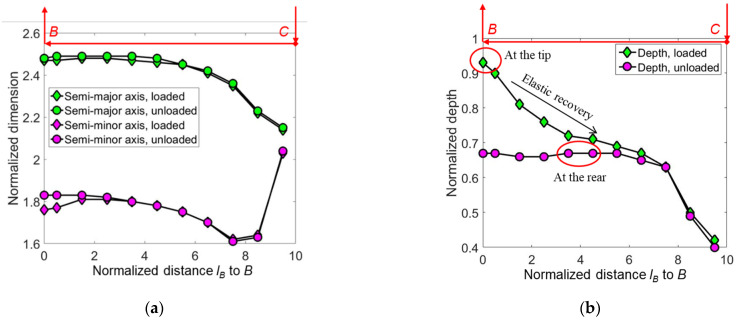
The fitted (**a**) dimension and (**b**) depth of the elastic–plastic boundary normalized by *h_max_*. The indenter moves from the point *C* to the point *B* in the scratching stage (see Figure 2).

**Figure 16 micromachines-13-01106-f016:**
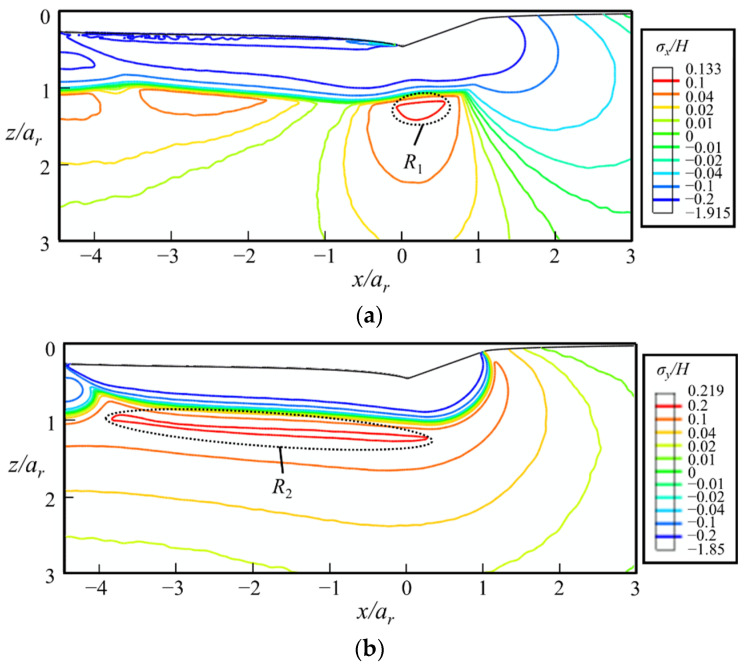
The contours of (**a**) *σ_x_*/*H*, (**b**) *σ_y_*/*H,* and (**c**) *σ_z_*/*H* at the end of the sliding stage ② in the *xz*-cross-section, where *a_r_* = 2.24 *h_max_* is the real contact radius evaluated by FEA, and *H* is the hardness measured by indentation tests. The indenter moves along the positive direction of the *x*-axis.

**Figure 17 micromachines-13-01106-f017:**
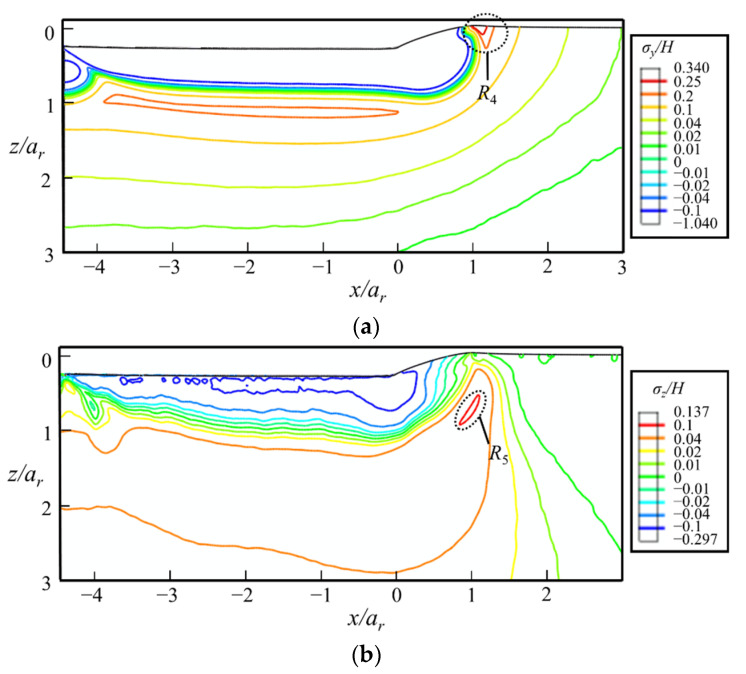
The contours of (**a**) *σ_y_*/*H* and (**b**) *σ_z_*/*H* at the fully unloaded state in the *xz*-cross-section predicted by FEA. The indenter moves along the positive direction of *x*-axis.

**Figure 18 micromachines-13-01106-f018:**
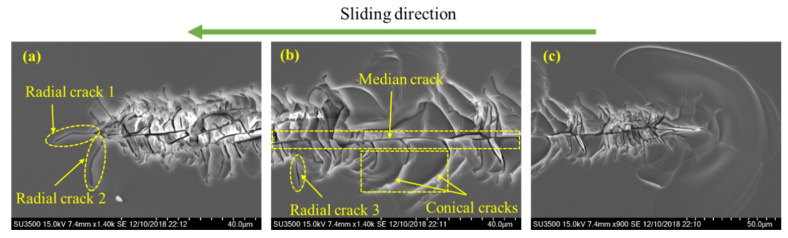
The SEM images of the impression at the (**a**) end, (**b**) middle, and (**c**) start of the edge-forward Berkovich scratching, under the normal load of 600 mN.

**Figure 19 micromachines-13-01106-f019:**
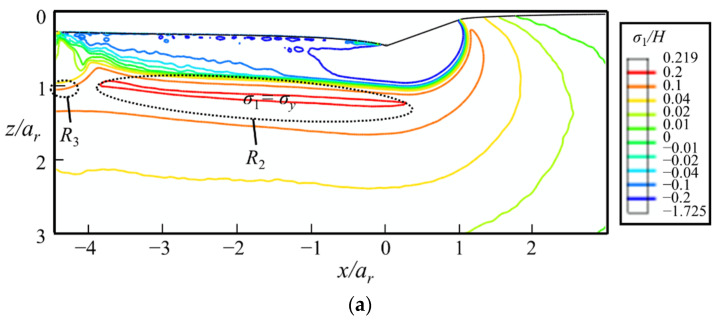
The contours of *σ*_1_/*H* at the (**a**) fully loaded and (**b**) fully unloaded states in the *xz*-cross-section. The thick dash line is the boundary between *σ*_1_ ≈ *σ_y_* and *σ*_1_ ≈ *σ**, where *σ** is the in-plane principal stress.

## Data Availability

Some or all data, models, or codes generated or used during the study are available from the corresponding author by request.

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
