# Peer review of "An Investigation into the Densification-Affected Deformation and Fracture in Fused Silica under Contact Sliding"

_micromachines, 2022, doi:10.3390/mi13071106_

Round 1

Reviewer 1 Report

Overall, this is a very nice scratch analysis simulation/experimental paper. I have a few notes:

 (1) Some details of the finite element model are not clear. Did you assume an infinitely sharp indenter? What are the materials properties of the indenter?

 (2) Also, when you try to verify the finite element model, why the elastic recovery ratio is used? Are there other ways to validate the model? For instance, matching forces?

 (3) In Figure 5 the residual depth is ~ 550 nm while it is mentioned 668 nm in the text (page 5).

Reviewer 2 Report

The topic of the work is relevant because it affects the issue of manufacturing optical materials, which are widely used both in the development of research and industrial production equipment.

1. The abstract almost completely repeats the conclusions.

2. The first section of the work provides a brief literature review describing the current state of research in the field of processing optical materials (glasses).

To review and discuss the results, various literature is used, including 10 publications over the past 5 years.

3. The second section describes the experimental part of the work - scratch tests. The description of the method is detailed and accompanied by explanatory illustrations.

4. The third section describes the finite element modeling technique.

The methods chosen in the second and third sections are generally accepted and correspond to the purpose of the work.

5. The fourth section presents the results of verification of the developed mathematical model. The simulation results are verified experimentally using scratch tests and atomic force microscopy.

6. The fifth section presents the results of studying the deformation and destruction of samples as a result of the impact of the indenter. Both the results of experiments and simulation based on the developed new mathematical model are presented.

The description of the results presented in the fourth and fifth sections of the article is accompanied by a discussion of them. At the same time, both their own reasoning and their analysis are given in conjunction with known literature data.

7. The sixth section presents the conclusions drawn from the new results obtained.

Notes:

1. The abstract should reflect the originality and novelty of the research, reflect the essence of the work and the main content of the article. Now the abstract is more like conclusions.

2. The conclusions are written in too general terms. Not enough specifics about the new results obtained are given.
